# Chemical Composition and Antimicrobial Activity against Phytopathogenic Fungi of Essential Oils Obtained from *Echinophora tenuifolia* subsp. *sibthorpiana* Grown in Wild and Cultivated Conditions in Turkey

**DOI:** 10.3390/molecules28020585

**Published:** 2023-01-06

**Authors:** Arif Sanli, Fatma Zehra Ok

**Affiliations:** Department of Field Crops, Faculty of Agriculture, Isparta University of Applied Science, 32200 Isparta, Turkey

**Keywords:** *E. tenuifolia* subsp. *sibthorpiana*, essential oil components, cultivation, antifungal activity

## Abstract

The hydro-distilled essential oils obtained from aerial parts of the wild and cultivated *Echinophora tenuifolia* subsp. *sibthorpiana* have been analyzed by GC-MS and screened for antimicrobial activity. In total, 28 compounds representing more than 99% of the oils were identified. α-phellandrene (13.22% and 55.27%), δ-3-carene (49.29% and 4.03%), and methyl eugenol (22.59–25.69%) were found as the main components for the wild and cultivated oils, respectively. EOs of the wild and cultivated plants differed significantly in both the percentage of the main components and antifungal effect. α-phellandrene was more dominant in cultivated plants (55.27%) than in wild ones (13.22%), while δ-3-carene was more abundant in the wild plants (49.29%). In the antifungal assays, both oils displayed moderate to high activity against three phytopathogenic fungi; *Fusarium oxysporum*, *Rhizoctonia solani,* and *Alternaria alternata*.

## 1. Introduction

Turkey is regarded as an important gene-center for Apiaceae family. The family Apiaceae is represented by 101 genera belonging to 485 species included in 511 taxa comprising 181 endemics in Turkey. The ratio of species endemism in the family is 37.3% [1]. The genus Echinophora (Apiaceae), which consists of about 10 species, is found in a very large area extending from the Mediterranean to as far as Afghanistan [2]; there are six species in the flora of Turkey, of which three are endemic. *Echinophora tenuifolia* L. subsp. *sibthorpiana* (Guss.)*,* a greyish-pubescent perennial herb, with height of 20–50 cm, and yellow petals, is one of the most significant species of this genus [3], and it is locally known as “çörtük” and “tarhana otu” [1]. The plant is a traditional ingredient to cheeses, tomato paste, and pickles as a spice for its desirable flavor and antimicrobial qualities. In addition, it is used as a treatment for injuries, gastric ulcers, and as a digestive remedy in traditional medicine due to its antifungal, carminative, and digestive traits [4]. It is also added to meals, such as soup, meat, pickles, dairy products, and meatballs, to enhance their taste properties [5,6]. Recently, the interest in medicinal and aromatic plants and the products obtained from them, which have been used in different fields, has been increasing. The usage of essential oils (EOs) as natural insecticides, antioxidants, and antibacterial agents, besides their use in the development of new products in pharmacology, cosmetics, and food industries, is rising by the day. Various plant EOs were shown to exert biological activity and are used as bio-pesticidal compounds [7,8]. Results of previous studies indicated that the antifungal activities of the extracts and EOs from some Echinophora species against phytopathogenic fungi [9,10,11,12,13].

Turkey is one of the countries where the production of medicinal plants is based on harvesting in the wild. The wild population of EO plants collected from the natural vegetation is under threat because of the strong fascination and demand, which has made numerous species scarce and sporadic due to over-utilization and aridity; there also a lack of protection measures for threatened therapeutic plant species [14]. In addition, since wild plants are not genetically homogeneous, problems with quality and standardization occur in the collected products.

Cultivation of these medicinal plants might become a promising alternative for their conservation and sustainable use. Several studies have already shown the interest of culture practice on the conservation of medicinal plants [15,16,17]. Cultivating a wildly grown plant under field conditions can affect the growth, content, and composition of secondary metabolites. Therefore, it is wise to evaluate the domestication effect of wild medicinal plants, in lieu of using wild gathered plant, on plant growth, EOs composition, and biological traits, before any field farming program. To our knowledge, there is no reports on biological activities of *E. tenuifolia* L. subsp. *sibthorpiana* (Guss.) EOs and on the comparison of plants grown in natural flora and culture forms. Thus, the aim of this study was to determine the chemical composition and evaluate the antifungal activities of *E. tenuifolia* L. subsp. *sibthorpiana* (Guss.) EOs and investigate the effect of cultivation on EO composition and biological activities.

## 2. Results and Discussions

### 2.1. EO Yield and Chemical Composition

Fresh and dry weights of plants grown under culture conditions were found to be more than four times higher than wild plants. The contents of EOs obtained from aerial parts were 1.78 ± 0.34% for wild plants and 1.52 ± 0.13% for cultivated plants on a dry weight basis (*v*/*w*). Cultivation did not greatly affect EO production. However, EO yields of cultivated plants were more than 3.5 times higher than wild plants (Table 1).

Quantitative analyses of the chemical composition of the investigated EOs are shown in Table 2. Twenty-eight compounds were identified, representing more than 99% in both wild and cultivated plants but with great differences in their abundances. EOs of wild and cultivated plants were characterized by a large amount of monoterpene hydrocarbons (75.71 and 73.44%, respectively). EOs of wild plants contained high amount of δ-3-carene (49.29%), while the main component of cultivated plant’s EOs was determined as α-phellandrene (55.27%). The phenylpropanoid derivative methyl eugenol was also present in the wild and cultivated EOs with a high percentage (22.59 and 25.69%, respectively). Other major components identified in oils from wild and cultivated plants were β-myrcene (3.23–0.86%), p-cymene (0.86–2.67%), limonene (2.03–1.63%), β-phellandrene (0.59–6.85%), and α-terpinolene (2.25–0.24%) (Table 2).

Cultivation did not greatly affect EO production. It is thought that the partial decrease in the EO content in the cultivated plants is due to agricultural practice such as irrigation and fertilization. It was reported that the essential oil content of *E. tenuifolia* subsp. *sibthorpiana* ranged between 1–2% and 0.76–2.4% [18,19,20] from various Turkish collections. Our results are similar to those reported by Şanlı et al. [19] and Başer et al. [20], who reported that the essential oil content of plants harvested at the full flowering stage was 2.0% and 1.76%, respectively. EO composition of *E. tenuifolia* subsp. *sibthorpiana* is similar to the composition of other *E. tenuifolia* subsp. *sibthorpiana* samples collected from Greek and Iran origins, and in different locations in Turkey. Some reports on the EO composition of *E. tenuifolia* subsp. *sibthorpiana* from various locations were previously published. The monoterpenes α-phellandrene (amounting to 51.5%) and δ-3-carene (amounting to 36.6%), and the phenylpropanoid derivative methyl eugenol (amounting to 80.6%), were reported by most of the authors as the main constituents of *E. tenuifolia* subsp. *sibthorpiana* EO [18,21,22]. Akgul and Chialva [23] reported the main compounds as a-phellandrene (51%), methyl eugenol (25%), δ-3-carene (5.7%), b-phellandrene (5%), and p-cymene (4.3%). Şanlı et al. [19] found the major components to be α-phellandrene (47.43–66.39%) and methyl eugenol (21.29–38.72%). Baser et al. [20] also reported α -phellandrene, p-cymene, and methyl eugenol as the principal compounds. Georgiou et al. [3] found that the EO contained 28 compounds and the main components of the oil were α-phellandrene (43.8%) and methyl eugenol (28.6%). Chalchat et al. [18] did not indicate α-phellandrene, but δ-3-carene and methyl eugenol were among the main constituents of Turkish origin. Ozcan et al. [22] determined methyl eugenol, δ-3-carene, p-cymene, and α-phellandrene in the EO from plants growing in Turkey. In another study, the main constituents of the EO were α-phellandrene (51.52%), methyl eugenol (17.46%), and p-cymene (14.66%) [20]. Telci and Hisil [6] reported 45 components in EO of the same species and methyl eugenol (52.4–62.9%), α-phellandrene (30.4%), p-cymene (7.8–9.1%), and δ-3-carene (3.3–5.7%) were the main components of the oil. It is probable that the geographic area of the plant collection and the vegetation stage of the plant are among the factors that effect on the oil yield and the composition of this subspecies. In addition, *E. tenuifolia* subsp. *sibthorpiana* populations may have different chemo types.

### 2.2. Antifungal Activity

The inhibitory effects of wild and cultivated *E. tenuifolia* subsp. *sibthorpiana* EOs were evaluated against three phytopathogenic fungi, *Fusarium oxysporum*, *Rhizoctonia solani,* and *Alternaria alternata*. Our results indicated that in comparison with the control, all concentrations of the tested oil could significantly inhibit fungal growth and exhibited a moderate to high antifungal activity. Both EOs showed a strong antifungal activity on *R. solani* and mean growth inhibition percentage was over 80% for this pathogen. The antifungal effects of wild and cultivated EOs on *F. oxysporum* and *A. alternata* were moderate, and the mean growth inhibition percentages in these pathogens were 41.3–55.3% and 45.1–48.4%, respectively. EOs form cultivated plants exhibited the higher inhibitory effect against *F. oxysporum* and *R. solani* than wild plant EOs. The inhibitory effect of both EOs on *A. alternata* was similar. Antifungal activity of the EOs increased with the increase in application doses for all three phytopathogens, and the highest antifungal activity was determined at 1500 and 2000 ppm concentrations, without significant differences between them. The interaction of EO × dose was found to be significant for the *A. alternata*, while the cultivated plant EO at a dose of 2000 ppm was more effective; there was no significant difference between cultivated and wild EOs at other doses (Table 3).

Because of the notable surge and significance of fungal diseases, which are hard to treat, it became imperative to find new solutions other than synthetic chemical fungicide [24]. Plant EOs are promising source of antifungal compounds. Several studies on plant pathogenic fungi have shown that some EOs have antifungal properties that inhibit the effect of fungi growth [25]. This can be associated with their varying make ups, including physical formation of the components and their functional groups and the likely cooperative relationships among the constituents [26]. There are very few literature data that Eos from Echinophora species are tested for antifungal activity. Eos of *Echinophora tenuifolia* subsp. *sibthorpiana* showed moderate antifungal activity against *Alternaria alternata, Aspergillus niger,* and *Aspergillus parasiticus,* and fungitoxic activity of the oil increased with increasing concentration [10]. The antimicrobial effects of the extracts and EO of *Echinophora sibthorpiana* Guss. against human pathogenic strains found to be similar to the positive controls, streptomycin and fluconazole [13]. It has been reported that *Echinophora spinosa* EO, which contains high content of δ-3-carene (60.86%), has high antimicrobial activity to human pathogens [27]. Ethanol extracts of *Echinophora platyloba* are tested for antimicrobial activity against *Candida albican*, *Staphylococcus aureus*, *S. epidermidis,* and *Streptococcus pyogenes* and showed weak antibacterial but potent antifungal activity [9,28]. Methanol extract of *E. platyloba* showed good antibacterial activity against *Staphylococcus aureus* and *Pseudomonas aeruginosa*, while there was no considerable growth prevention for *Aspergillus flavus*, *A. niger,* and *Candidia albicans* [29].

The EO of *E. tenuifolia* subsp. *sibthorpiana* has shown moderate to high degree of growth inhibition effect against tested fungal phytopathogens. It is thought that the main constituents (α-phellandrene, δ-3-carene, and methyl eugenol) to be responsible for the growth inhibition of pathogenic fungi. Moreover, the remaining frictional substance in the oil is believed to play as a cooperative role as it has been proposed [30]. Both wild and cultivated *E. tenuifolia* subsp. *sibthorpiana* EOs seem to have a similar robust antifungal activity when compared against each other. The reason why cultivated plants is more effective than wild plants on *F. oxysporum*, *R. solani* can be contributed to the variance of the major compounds that occurs in both cultures. Recently, numerous scientists have reported that monoterpene hydrocarbons and their oxygenated extracts and ingredients, which have frequently been used as biological agents for their healing properties, have huge potential to inhibit bacterial pathogens and toxicity against plant pathogenic fungi [31]. In general, the antifungal activity of the EO is mostly due to the presence of phenols such as methyl eugenol [32] and monoterpene hydrocarbons such as α-phellandrene [33,34] and δ-3-carene [35]. The bioactivity of these compounds has been correlated with their lipophilic character that gives them the ability to penetrate cell walls. Therefore, the monoterpene hydrocarbons and phenols had important role in antifungal properties of *E. tenuifolia* subsp. *sibthorpiana*. The various outcomes gained when used different concentrations can be linked with the arrangement of the hydrogen bonds between the hydroxyl group of oil phenolics [36] and the monoterpenes that are found in the oil, which amplifies the concentration of lipidic peroxides and consequently results in cell death [37]. The significant role that components have in inhibiting fungal activity is due to the decline or physical changes that occur in high phenols such as methyl eugenol and the probable useful harmonious exchanges between them [36].

## 3. Materials and Methods

### 3.1. General

This study was carried out using wildly grown and cultivated *E. tenuifolia* subsp. *sibthorpiana*. The seeds of plants that are morphologically similar to each other from wild plants were collected in Lakes Region of Turkey at full maturity in September 2019. After surface sterilization of the seeds, they were planted in viols containing 3:1 peat-perlite mixture in November 2019 and left to germinate in open field conditions. The seedlings developed in the viols were planted in four rows of 10 m in length, 1.2 m in wide, and 50 cm spacing within each row, in April 2020. For the first month, the crop was watered twice a week. For the subsequent months, irrigation was carried with an interval of 15 or 21 days, as needed, with a drip irrigation system. Weed control was done manually depending on the weed density. At the beginning of each vegetation year, 10 kg/da nitrogen and 8 kg/da phosphorus were applied with irrigation systems.

The experimental area is located in the Faculty of Agriculture, Isparta University of Applied Sciences in Isparta Province (37°45′ N, 30°33′ E, elevation 1035 m) in Turkey. The soil texture of the experimental area was clay loam with a pH of 8.2. Lime content of the soil was 7.1% (Scheibler calcimeter), organic matter content was 1.3% (Walcley–Black method), and the total salt was 0.29%. Total nitrogen content of experimental area was 0.29% (Kjelhdal method), extractable phosphorus and exchangeable potassium contents were 16.7 mg kg^−1^ by 0.4 N NaHCO3 extraction and 179 mg kg^−1^ by 1 N NH4OAc, respectively, and available sulfate was 17.3 mg kg^−1^. The experimental area is characterized by a semi-arid bioclimate with a mean rainfall of 498 mm/year and an annual average temperature of 12.1 °C.

The aerial parts of the cultivated and wild plants were harvested (five plant samples with three replications) at the flowering stage at the end of July 2021. Plant materials were identified by Dr. Hasan ÖZÇELİK according to “Flora of Turkey” [38] and voucher specimens (63.69.11.1-5) were deposited in the Herbarium GUL, Suleyman Demirel University. The fresh herb weight was determined by weighing the herbs of five plants harvested from the wild and cultivated plants, and the dry herb weight was calculated by drying the herbs in the shade. EO yields were calculated by using the EO ratios and dry herb weights.

### 3.2. Essential Oil Extraction

Dried plants (1 kg) and distilled water (3 L) were placed in a flask (6 L) connected to the condenser of a Clevenger type hydro-distillation apparatus (Calıskanlab, Ankara, Türkiye) according to standard procedure as described in the European Pharmacopoeia [39]. After 3 h of distillation, the EO and the water mixture were finally separated by decantation. EO yield was measured as a percentage (*v*/*w*) on average in triplicate analyses. The EO were dried with anhydrous sodium sulphate and stored at 4 °C until used for analysis to determine the EO compounds by GC-FID/GC-MS analysis.

### 3.3. GC-FID/GC-MS Analysis

Gas Chromatography/Mass Spectrometry (GC-MS) analysis of the EOs (50 μL of the oil was solubilized in 5 mL of n-hexane and injected into the split mode 1/100) was performed on Shimadzu 2010 Plus GC-MS (Kyoto, Japan) equipped with a Quadrapole (QP-5050) detector. The analysis was employed under the following conditions: capillary column, CP-Wax 52 CB (50 m × 0.32 mm, film thickness 0.25 μm); injector and detector heats, 240 °C; stove heat program, from 60 °C (10 min. hold) to 90 °C rising at 4 °C/min, and increasing to 240 °C (11.5 min hold) rising at 15 °C/min.; flow speed, 1 psi; detector: 70 eV; ionization type, EI; carrier gas, helium (20 mL/min); sample injected, 1 µL. Identification of constituents was carried out with the help of retention times of standard substances by composition of mass spectra with the data given in the Wiley, Nist, Tutor library [40,41]. The quantitative analysis was conducted using Gas Chromatography/Flame Ionization Detector (GC-FID), Shimadzu Model Thermo Ultra Trace, operating at the same conditions of GC-MS.

### 3.4. Fungi Cultures

Plant pathogens (*R. solani*, *F. oxysporum,* and *A. alternata*) were obtained from stock cultures at Department of Plant Protection, of Isparta University Applied Sciences. Fungi cultures were developed at 20 mL potato dextrose agar (PDA) on petri dish (90 mm) and kept at 22 ± 2 °C for 7 days and these fungi were used for the experiment.

### 3.5. In Vitro Antifungal Activity and Fungal Growth Inhibition of the EOs

The antifungal activities were determined by using agar plate methods [42]. PDA [95 mL (*w*/*v*)] was autoclaved and kept at 40 °C. For preliminary assessment and screening the efficacy of EOs obtained from wild and cultivated plants, the effect of both EOs at 500, 1000, 1500, and 2000 ppm concentration on growth of the pathogens was investigated. PDA media treated with the EOs were prepared by adding appropriate quantity of both EOs to melted medium, followed by addition of Tween 80 (0.01%) to disperse the EO in the medium. The mycelium disc (10 mm in diameter) from 7-day-olds fungi cultures were transferred to petri plates (60 mm). Following incubation at 28 °C for 72 h, growth of fungi was recorded daily [43]. The positive control (without EO) plates were inoculated following the same procedure. Tween-80 (0.1%) was used as a negative control. The growth inhibition percentage was calculated according to the formula described by [44], which is mentioned below:Growth inhibition (%) = (C − T)/C) × 100
where, C is the diameter of mycelial growth in control plates, and T is the diameter of mycelial growth in treated plates. Three replicates were used per treatment, and the experiment was repeated three times.

### 3.6. Experimental Design and Data Analysis

The data obtained from the research were subjected to analysis of variance (ANOVA) using a completely randomized design. The significance test for comparing the differences among the means was carried out with LSD (*p* < 0.05) using SAS (2009) statistical software.

## 4. Conclusions

To the best of our knowledge, the composition and biological activities of wild and cultivated *E. tenuifolia* subsp. *sibthorpiana* EOs are reported for the first time. Our experiments illustrated that the *E. tenuifolia* subsp. *sibthorpiana* had a high yield potential under culture conditions and more than 3.5 times more essential oil was obtained from the unit area compared to wild plants. Although essential oil components were almost the same under culture conditions, significant changes occurred in their amounts. In particular, δ-3-carene, which is the main component of wild EOs, was replaced by α-phellandrene in the culture conditions. Our results highlighted that the EO of *E. tenuifolia* subsp. *sibthorpiana* is effective for inhibition or control of *F. oxysporum*, *R. solani,* and *A. alternata,* and that EOs from the cultivated plants showed higher biological activity with respect to the wild plants. Therefore, it can be summarized that higher EO production and biological effectiveness can be achieved by cultivating this plant.

## Figures and Tables

**Table 1 molecules-28-00585-t001:** Fresh and dry herb weights, EO ratios, and yields of wild and cultivated *E. tenuifolia* subsp. *sibthorpiana*.

Plant Samples	Fresh Herb (Weight/g/Plant)	Dry Herb Weight (g/Plant)	Essential Oil Content (%)	Essential Oil Yield (mL/Plant)
Wild	342 ± 27.4	130 ± 12.0	1.78 ± 0.11	2.33 ± 0.34
Culture	1738 ± 101.2	548 ± 29.6	1.52 ± 0.13	8.33 ± 0.37

**Table 2 molecules-28-00585-t002:** Chemical composition of the essential oils of wild and cultivated *E. tenuifolia* subsp. *sibthorpiana*.

No	RI^a^	RI^b^	Compounds	Wild (%)	Cultivated (%)
1	927	928	α-thujene	0.26	0.45
2	933	936	α-pinene	0.67	0.73
3	943	946	Camphene	0.02	Tr
4	967	969	Sabinene	0.18	0.22
5	976	977	β-pinene	0.04	0.13
6	985	988	**β-myrcene**	**3.23**	**0.86**
7	1001	1002	**α-phellandrene**	**13.22**	**55.27**
8	1009	1008	**δ-3-carene**	**49.29**	**4.03**
9	1018	1018	α-terpinene	0.26	0.32
10	1025	1027	**p-cymene**	**0.86**	**2.67**
11	1030	1032	**Limonene**	**2.03**	**1.63**
12	1031	1033	**β-phellandrene**	**0.59**	**6.85**
13	1036	1034	1,8-cineole	0.04	Tr
14	1041	1040	β-ocimene, (Z)	1.19	0.05
15	1047	1049	β-ocimene, (E)	0.76	0.00
16	1054	1050	γ-terpinene	0.82	0.31
17	1072	1068	Sabinene hydrate (E)	0.79	0.02
18	1086	1086	**α-terpinolene**	**2.25**	**0.24**
19	1106	1106	β-pinene oxide	0.44	0.18
20	1125	1128	p-mentha-1,5,8-triene	0.06	Tr
21	1135	1130	Neo-allo-ocimene	0.30	Tr
22	1143	1140	Epoxy-ocimene (E)	0.06	Tr
23	1184	1187	2-Dodecen-4-yne, (E)	0.05	Tr
24	1186	1188	Sabinol (E)	Tr	0.10
25	1197	1198	p-Mentha-1(7),8-dien-2-ol (E)	Tr	0.03
26	1280	1287	Sabinyl acetate	Tr	0.07
27	1286	1288	Limonene dioxide 1	Tr	0.04
28	1403	1403	**Methyl eugenol**	**22.59**	**25.69**
			Monoterpene hydrocarbons	75.71	73.44
			Oxygenated monoterpenes	1.53	0.52
			Phenolic components	22.59	25.69
			Others	0.17	0.24
			Total	100	99.89

RI^a^: Retention indices of each compound calculated by retention time with that of n-alkanes; RI^b^: Retention indices that refers to NIST Chemical Web Book; Tr: Traces.

**Table 3 molecules-28-00585-t003:** Antifungal activity of wild and cultivated *E. tenuifolia* subsp. *sibthorpiana* EOs.

		Percent Inhibition of Mycelia Growth			
		Concentrations (ppm)			
Pathogens		500	1000	1500	2000
	Wild	29.2 ± 8.2f	41.6 ± 7.8cde	48.9 ± 7.3bc	45.5 ± 5.0cd
*F. oxysporum*	Cultivated	37.3 ± 8.1def	55.8 ± 4.5ab	64.1 ± 9.0a	63.7 ± 3.4a
CV (%): 14.4					
	Wild	73.1 ± 0.9e	78.4 ± 0.4d	80.3 ± 0.9cd	81.4 ± 1.3bc
*R. solani*	Cultivated	74.8 ± 1.4e	81.1 ± 1.2bc	83.4 ± 0.4ab	84.5 ± 1.2a
CV (%): 1.30					
	Wild	28.3 ± 7.1c	50.8 ± 8.6b	50.4 ± 3.5b	50.8 ± 5.2b
*A. alternata*	Cultivated	33.3 ± 0.8c	46.6 ± 0.9b	52.1 ± 0.4b	61.4 ± 3.3a
CV (%): 10.0					

The results are means ± standard errors of four replications. Means within a column indicated by the same letter were not significantly different according to LSD test at the level *p* < 005. CV: Coefficient variance.

## Data Availability

Not applicable.

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
