# Peer review of "Chemical Composition and Antimicrobial Activity against Phytopathogenic Fungi of Essential Oils Obtained from Echinophora tenuifolia subsp. sibthorpiana Grown in Wild and Cultivated Conditions in Turkey"

_molecules, 2023, doi:10.3390/molecules28020585_

Round 1
Reviewer 1 Report
The article presents chemical composition and antifungal activity of essential oils obtained from aerial parts of the wild and cultivated Echinophora tenuifolia subsp. sibthorpiana. According to the Authors, the novelty of the publication is recognition of the biological activity of these essential oils, regarding plants growing in the natural environment and those cultivated.
11. However, articles on the antifungal activity of E. tenuifolia EO have been previously published (e.g. Coskun Saglam, Mehmet Musa Özcan, Nuh Boyraz (2009) Fungal Inhibition by Some Spice Essential Oils. Journal of Essential Oil Bearing Plants, 12(6), 742-750; Mileski K., Džamić A., Ćirić A., Grujić S., Ristić M., Matevski V., Marin P.D. (2014) Radical Scavenging and Antimicrobial Activity of Essential Oil and Extracts of Echinophora sibthorpiana Guss. from Macedonia. Arch. Biol. Sci., 66(1), 401-413) and should be included in the discussion.
22. The title of the article suggests a broader study of antifungal activity, while this activity was determined against 3 phytopathogenic moulds. This scope of research should be emphasized in the title, e.g. “Chemical Composition and Antimicrobial Activity against Phytopathogenic Fungi of Essential Oils Obtained from Echinophora tenuifolia subsp. sibthorpiana Grown in Wild and Cultivated Conditions in Turkey”.
33. In the botanical nomenclature, it is recommended, besides the name of the genus and the species name, to provide the name of the botanist who originally described the species, i.e. Echinophora tenuifolia L. subsp. sibthorpiana (Guss.). It is worth adding this information in the introduction.
44. Abstract (lines 14-16): “The volatiles of the wild and cultivated material differed significantly in both the percentage of the main components and antifungal effect.” - The methodology indicates that the antifungal activity was determined for essential oils, not their volatile fractions. Please correct.
55. Abstract (lines 17-19): “In the antifungal assays, both oils displayed moderate to high activity against all phytopathogens tested with the oil of the cultivated form being more active.” – “all phytopathogens” is too general. Please list the mould species tested.
66. Table 2 – Please complete description of the table, i.e. what data is presented (content?), what unit.
77. The correct name of the species is Fusarium oxysporum not oxysporium. Please change everywhere in the text. Please, check: http://www.indexfungorum.org/names/Names.asp
88. Line 118: “…evaluated against three phytopathogenic fungi, F. oxysporium, R. solani and A. alternata”. Names of species mentioned for the first time in the text should be given in full.
99. Table 3 – as in table 2 please add units and more detailed description. What do the letters A, B and C mean? What does the abbreviation CV mean?
110. Only averages are given in table 3. Please add standard deviation values.
111. It is not clear to me why the Authors calculate the average of the growth inhibition values for different concentrations of EOs (table 3). After all, it is known that the activity is strictly dependent on their concentrations. What is the purpose of averaging? Please include this comment in the description of the results and the discussion.
112. Similarly, it is unreasonable to calculate the average for EOs from wild and cultivated plants when the origin of the plants is the key difference discussed (table 3). Please include this comment in the description of the results and the discussion.
113. Line 146: please correct Streptococcus pyogenes.
114. Lines 151-152: “It is thought that the main constituents to be responsible for the growth inhibition of pathogenic fungi.” What constituents?
115. Lines 155-157: “The reason why cultivated F. oxysporium, R. solani is more effective than wild plants can be contributed to the variance of the major compounds that occurs in both cultures”. The meaning of this sentence is unclear. Please, correct.
116. Line 165: “walls 41”. Is this a citation?
117. Lines 170-173: “The significant role that components have in inhibiting fungal activity is due to the decline or physical changes that occur in high phenols such as methyl eugenol and the probable useful harmonious exchanges between them as stated in this study [29].” It is unclear. Please correct.
118. Lines 173-174: “These results were consistent with those of previous studies describing the antifungal activity of these volatile compounds [30, 31]”. Again, the Authors did not test the antifungal activity of the EOs’ volatile fractions.
Author Response
Dear reviewer,
Necessary corrections have been made in line with your suggestions. Please see the attachment in the box.

Reviewer 2 Report
Dear authors,
This study is interesting because it explores a comparation of chemical composition and antifungal activities of esential oil obtained from E. tenuifolia obtained in different condition.
Major comments
- The introduction should be improved, seven references are too few;
Minor comments
-give information about the equipment used-company, country
Author Response
Dear reviewer,
- Introduction section is improved and 5 more literature about the subject has been added,
- Necessary information about clevenger apparatus and GC-MS devices are given.
Reviewer 3 Report
Molecules-2120216_Comments 15/12/2022
Title: Comparison of Chemical Composition and Antifungal Activities of Essential Oil Obtained from Echinophora tenuifolia Subs. sibthorpiana Grown in Wild and Cultivated Conditions in Turkey
Dear Sir, the work is very interesting, scientific and with appropriate information. Therefore, the manuscript can be acceptable for publication after minor revision.
Comment list:
1) What about the materials and methods section, give the all information in separate heading using materials and methods?
2) Conclusion section should be revised more critical.

Author Response
Dear reviewer,
- All information has already been given under separate headings in the materials and methods section. Since the cultivated plants used as material include field production, material and method parts could not given as separate headings.
- Conclusion section has been revised in more critical.
Thank you very much for your contribution...
Round 2
Reviewer 2 Report
I believe this paper has merit to be published in the journal.